# Signal Preprocessing in Instrument-Based Electronic Noses Leads to Parsimonious Predictive Models: Application to Olive Oil Quality Control

**DOI:** 10.3390/s25030737

**Published:** 2025-01-25

**Authors:** Luis Fernandez, Sergio Oller-Moreno, Jordi Fonollosa, Rocío Garrido-Delgado, Lourdes Arce, Andrés Martín-Gómez, Santiago Marco, Antonio Pardo

**Affiliations:** 1Department of Electronics and Biomedical Engineering, Universitat de Barcelona, 08028 Barcelona, Spain; lfernandez@ub.edu (L.F.); santiago.marco@ub.edu (S.M.); 2Institute for Bioengineering of Catalonia (IBEC), The Barcelona Institute of Science and Technology, 08028 Barcelona, Spain; 3B2SLab, Departament d’Enginyeria de Sistemes, Automàtica i Informàtica Industrial, Universitat Politècnica de Catalunya, 08034 Barcelona, Spain; jordi.fonollosa.m@upc.edu; 4Institut de Recerca Sant Joan de Déu (IRSJD), 08950 Esplugues de Llobregat, Spain; 5Networking Biomedical Research Centre in the Subject Area of Bioengineering, Biomaterials and Nanomedicine (CIBER-BBN), 28029 Madrid, Spain; 6Department of Analytical Chemistry, University of Córdoba, 14071 Córdoba, Spain; rogade1983@gmail.com (R.G.-D.); qa1arjil@uco.es (L.A.); q02magoa@uco.es (A.M.-G.)

**Keywords:** MCC-IMS-based e-nose, signal processing workflow, preprocessing, parsimony, validation, olive oil

## Abstract

Gas sensor-based electronic noses (e-noses) have gained considerable attention over the past thirty years, leading to the publication of numerous research studies focused on both the development of these instruments and their various applications. Nonetheless, the limited specificity of gas sensors, along with the common requirement for chemical identification, has led to the adaptation and incorporation of analytical chemistry instruments into the e-nose framework. Although instrument-based e-noses exhibit greater specificity to gasses than traditional ones, they still produce data that require correction in order to build reliable predictive models. In this work, we introduce the use of a multivariate signal processing workflow for datasets from a multi-capillary column ion mobility spectrometer-based e-nose. Adhering to the electronic nose philosophy, these workflows prioritized untargeted approaches, avoiding dependence on traditional peak integration techniques. A comprehensive validation process demonstrates that the application of this preprocessing strategy not only mitigates overfitting but also produces parsimonious models, where classification accuracy is maintained with simpler, more interpretable structures. This reduction in model complexity offers significant advantages, providing more efficient and robust models without compromising predictive performance. This strategy was successfully tested on an olive oil dataset, showcasing its capability to improve model parsimony and generalization performance.

## 1. Introduction

The electronic nose concept was born from the seminal work of Persaud and Dodd in 1982 based on the study of the odor discrimination capacity of the olfactory system of mammals [1]. In this study, Persaud and Dodd proposed that olfactory systems do not require highly specific detectors to achieve odor discrimination. They developed an electronic model of the mammalian olfactory system, utilizing non-specific gas sensors combined with a feature detection system. Later, in 1999, Gardner and Bartlett introduced the classical definition of an electronic nose as an instrument that comprises an array of non-specific electronic chemical sensors and a pattern recognition system [2]. Electronic noses have been applied in various sectors, each with specific applications. In the food and beverage sector, they have been successfully employed for quality control, food classification, freshness assessment, authenticity verification, and origin traceability. In environmental monitoring, they have demonstrated effectiveness in evaluating polluting gasses, analyzing air quality, identifying sources of malodors, and controlling industrial processes. Similarly, in the healthcare sector, electronic noses have shown promising potential for detecting diseases such as lung cancer, leukemia, schizophrenia, and diabetes, as well as for identifying pathogens like Salmonella Typhimurium and Escherichia coli. Beyond these applications, electronic noses are also being explored in sectors such as automotive, packaging, cosmetics, and security, among others [3,4,5,6,7,8,9]. However, despite the numerous applications, gas sensor-based e-noses face several limitations such as sensor poisoning, drift, lack of reproducibility, nonlinear response, sensitivity to humidity and temperature, limited selectivity, and detection limit in the ppm range. These limitations are also shared by emerging sensor technologies, such as those based on carbon nanotubes, which, despite their promise, still struggle with similar issues affecting reproducibility, sensitivity, and selectivity [10,11,12]. These challenges restrict the effectiveness of gas sensor-based e-noses in certain applications and have limited their widespread adoption in industry. To overcome the issue of low specificity in sensor-based e-noses, strategies involving the modulation of sensor operating parameters have been proposed. For example, modulating the operating temperature of metal oxide gas sensors improves their ability to distinguish between different gasses [13,14,15,16]. Indeed, the sensor’s response to a gas, when subjected to a slowly varying temperature, can be interpreted as a temperature-dependent spectrum, with different gasses producing distinct spectra. In this regard, modulating the operational parameters of sensors has blurred the line between traditional gas sensor-based e-noses and analytical chemistry instruments, expanding the e-nose concept to encompass alternative technologies. Ultra-fast gas chromatography (GC) coupled to mass spectrometry (MS) or to an ion mobility spectrometry (IMS) are examples of next-generation e-nose instruments commonly found in the literature, which also offer better selectivity and detection limits in the sub-ppm range, addressing applications that traditional gas sensor-based e-noses were unable to handle [17,18,19,20]. The challenges faced by gas sensor-based e-noses and instrument-based e-noses share several commonalities, such as noise reduction, baseline correction, and sensitivity to environmental factors like humidity, which, due to different physical mechanisms, significantly affects both [21,22,23,24]. However, a key distinction lies in how relevant information is distributed within the data. In conventional sensor-based e-noses, the relevant information is often dispersed across the entire response matrix of the sensor array. In contrast, in instrument-based e-noses, the relevant information is concentrated in discrete peaks within a spectral output. This difference presents both an opportunity and a challenge: while spectral peaks enable a more precise extraction of relevant information, they also demand specialized tools and algorithms for peak processing and spectral analysis. These unique requirements highlight the dual nature of instrument-based systems, which combine the principles of traditional gas sensors with the analytical precision of advanced spectroscopic methods.

Among the various instrument-based e-noses, those utilizing IMS stand out as particularly promising. IMS is an analytical technique that offers a rapid, robust, and highly sensitive methodology for gaseous phase chemical sample analysis. The instrument consists of three regions. The first is the ionization region, where the sample is ionized, and protonated ions are generated. Various ionization sources can be used, with the most common being a radioactive source. The second region is the drift tube region, where the ionic species are separated over time. Finally, there is the detection region, where the final spectrum is formed.

Briefly, if a radioactive source is used, the beta particles it emits interact with the supporting atmosphere in the IMS, forming a reservoir of reactant ions. These ions primarily consist of hydronium ions, H_3_O^+^(H_2_O)_n_, which constitute the so-called reactant ion peak (RIP). In the presence of a neutral volatile sample, the reservoir of reactant ions, rather than the beta particles, is utilized to generate protonated molecules through a two-step ionization process. The protonated molecules and remaining reactant ions are then introduced into the drift tube region via an electronic ion shutter, which controls their injection from the ionization chamber [22]. A large reservoir of ions is retained for subsequent scans under similar conditions [23,24]. Once in the drift tube, the ions travel toward the detector under a uniform electric field at atmospheric pressure, while interacting with a counter-flowing carrier gas. The ions are separated based on their electrical mobility, which depends on factors such as mass, charge, and collision cross-section among others. This mobility determines the time it takes for each ion to reach the detector. The resulting IMS spectrum consists of signal intensity versus drift time (or ion mobility). In the absence of analytes, only the RIP appears (it acts as an excess reagent), but when analytes are present, the RIP intensity decreases as analyte-specific peaks increase.

IMS technology is valuable for both qualitative applications, where substances are identified by their mobility, and quantitative applications, where substance concentrations are correlated with peak intensities. For a more detailed explanation of IMS functionality and a comprehensive list of applications, see references [25,26,27,28,29].

While these instruments perform well in various scenarios, they have limitations, particularly in moderate selectivity and nonlinear response. Analyzing complex gas mixtures, which may contain hundreds of compounds, often leads to peak overlapping for chemicals with similar mobilities. The limited supply of reactant ions also contributes to nonlinear behavior, as the instrument’s response saturates at higher analyte concentrations. Charge competition further limits the detection of analytes with lower proton affinities. Additionally, ion–molecule reactions within the drift tube can broaden reactant ion peaks, generating noisy spectra [27]. To address these limitations, IMS systems can be coupled with a multi-capillary chromatographic (MCC) column for fast chromatography with high sample flow rates or with a gas chromatography (GC) column. This combination functions like an analytical chemistry-based electronic nose, generating a unique electronic fingerprint of samples by integrating MCC with IMS. The MCC column first performs a pre-separation of volatile analytes before a second mobility-based separation occurs in the IMS as the analytes elute from the column. The orthogonal separation of an MCC-IMS e-nose enhances performance by reducing charge competition and improving selectivity. The MCC column separates interfering compounds before IMS ionization, resulting in more reliable spectra. With GC-IMS, a similar orthogonal separation is aimed at by employing a traditional gas chromatography column prior to IMS detection. Chromatographic resolution can improve due to the longer length of GC columns compared to MCC columns, which may facilitate the identification of molecules in complex mixtures. However, the longer length often results in longer analysis times, which can be a drawback for high-throughput applications that require rapid results. Furthermore, higher resolution might not always be necessary. Therefore, the choice depends on the application: MCC-IMS is best suited for faster analysis, although it comes with limited chromatographic resolution and increased data processing complexity.

The complexity of the generated data with both MCC-IMS and GC-IMS requires advanced processing and chemometric analysis techniques for its interpretation. The resulting raw data are presented as a two-dimensional signal intensity map. For each retention time, an ion mobility spectrum is captured, with specific ionic species forming distinct regions of interest on the 2D map. These regions represent the characteristic electronic fingerprint of the sample [28,29,30,31,32,33].

Hyphenated MCC-IMS or GC-IMS-based-e-nose instruments are able to face a wider range of opportunities than traditional gas sensor-based e-noses in several fields as food quality control [33,34], environmental monitoring, or diagnosis of diseases [35,36,37,38,39,40]. But, on the other hand, each 2D map provided by an MCC-IMS-based e-nose is composed of a large quantity of highly correlated data (over 10^6^ data points per sample), challenging data handling and processing [38,39]. In addition, for a set of 2D maps, instrumental shifts result in a different position and intensity of the located regions of interest corresponding to the same compound, making unfeasible the definition of fingerprints beforehand to correlate regions of the map with each compound of interest [40]. Therefore, in order to cope with collinearities, sample complexity, instrumental variability, and noise and matrix effects that cause spectral interferences, MCC-IMS-based e-nose data need to be preprocessed before they can be utilized to extract information from the measured chemical sample [30].

Some MCC-IMS preprocessing methodologies include specific steps to compensate for the RIP tail, which alters peak heights and hinders the quantification of analytes [40,41]. Even though most instruments already report the average of several spectra (typically 16 to 32), noise reduction strategies are still required. There are multiple possibilities to achieve noise reduction. Among them, the reconstruction of the signal by means of thresholding wavelet coefficients [40] or digital signal filtering [42] are two of the most widespread. Low-frequency noise may also be present in the spectra and cause slow baseline fluctuations. As a result, peaks containing analytical information appear superimposed on a smooth changing baseline. In order to obtain comparable peak intensities under similar conditions, baseline correction needs to be performed before further numerical processing [42,43]. Finally, peak alignment is essential to simplify ion tracking across different samples for a subsequent data analysis. The most used methods for this task are Parametric Time Warping (PTW), Interval Correlation Optimized Shifting (Icoshift) and Correlation Optimized Warping (COW) [44,45,46,47,48].

After data preprocessing, a feature extraction stage is generally conducted by identifying and integrating peaks within the 2D map of the analyzed sample [30,49]. This reduces the original MCC-IMS data to an array of intensities for each identified peak, characterized by their position in both retention and drift times (peak table approach). Alternatively, one can use all features from the 2D map (full-matrix approach). Although simpler, the full-matrix method skips peak picking, avoiding issues with peak area definition, especially for overlapping peaks. It also does not require predefined peak shapes, providing more flexibility during detection and integration. The final step consists of combining the features from each sample into a single matrix. This matrix can then be processed using common techniques such as Principal Component Analysis (PCA), k-nearest Neighbors (kNNs), Partial Least Square Discriminant Analysis (PLS-DA), Support Vector Machines, among others, which are selected according to their flexibility and complexity. To the best of our knowledge, only a few comprehensive preprocessing and feature extraction workflows for MCC-IMS data have been developed. For instance, Freire et al. introduced a MATLAB workflow to investigate the effect of various feature extraction methods on PLS-DA model classification accuracy [49]. In a more recent publication, Christmann and collaborators presented a Python library (GC-IMS-tools) that used a full-matrix approach for MCC-IMS data preprocessing [50]. Similarly, Oller et al. created an R package (GC-IMS) to process MCC-IMS data based on a peak table approach [51]. Interestingly, none of these studies evaluated the impact of preprocessing on the complexity of predictive models built from MCC-IMS data.

This paper introduces a methodology to refine chemical information from MCC-IMS-based e-nose data, utilizing a full-matrix approach and untargeted analysis. In the current paper, the process of transforming raw data into useful data for model building has been semi-automated. Our methodology refines MCC-IMS raw data through a comprehensive preprocessing workflow encompassing noise reduction, baseline correction, and peak alignment. This study systematically examines the influence of each preprocessing stage on the accuracy and effectiveness of the final classification. This new approach directly inputs the corrected 2D intensity maps for each sample into the model. We extract information on the sample from the processed matrix, such as latent variables in PLS-DA and quantify the generality of the models by exploring their accuracy in internal and external validation. We carry the complete matrix until the model is built, reducing the complexity of data preprocessing and feature extraction processes. In this study, we apply our methodology to process MCC-IMS data, using a challenging classification task to distinguish between three different classes of olive oil. Since each type of olive oil possesses unique organoleptic properties, they hold varying levels of market value. Therefore, the simple and accurate classification of olive oil types is very desirable for the industry to reduce costs and detect fraud [52]. The remainder of the paper is organized as follows. In Section 2, we briefly introduce the MCC-IMS-based e-nose used, and we describe the collected olive oil dataset used to illustrate our methodology. Also, in Section 2, we present the methodology to process MCC-IMS-based e-nose measured data, with specific details on denoising, baseline correction, peak alignment, multivariate classification, and model validation process. The performance of the applied methodology and the discussion of the results is carried out in Section 3. In particular, we emphasize how preprocessing influences the generalizability of PLS-DA models. Additionally, we detect which features in the dataset are responsible for class separation. Finally, Section 4 is dedicated to presenting the conclusions of this work.

## 2. Materials and Methods

### 2.1. Instrumentation and Dataset

We used a set of olive oil samples of three different categories to exemplify our methodology. Each sample was analyzed twice, resulting in a set of 180 experiments that corresponded to 3 different types of olive oil: 60 were obtained from “extra virgin” olive oil (EVOO) samples, 60 from “virgin” olive oil (VOO), and 60 from “lampante virgin” olive oil (LVOO).

The samples were stored at 4 °C in opaque glass containers and then analyzed by means of an MCC-IMS-based e-nose instrument (FlavourSpec^®^) from Gesellschaft für analytische Sensorysteme mbH (G.A.S.), Dortmund (Germany) according to EU Regulation 2568/91. The instrument was equipped with an autosampler device (CTC-PAL, CTC Analytics AG, Zwingen, Switzerland). An aliquot of 1 g of olive oil sample was placed in a 20 mL glass vial and it was hermetically closed with a magnetic cap. Then, each sample was heated at 60 °C for 10 min in order to generate a headspace in the vial. The selected temperature (60 °C) and incubation time (10 min) were chosen as they are sufficient to volatilize the aromatic compounds for analysis without significantly accelerating the oxidative or thermal degradation of the oil [53,54]. After this, 100 µL of the gaseous sample from the headspace was injected by the autosampler device into the heated splitless injector (80 °C) of the MCC-IMS instrument.

After injection, the gaseous sample was carried into a non-polar multi-capillary column (MCC OV-5, MultiChrom Ltd. Novosibirsk (Russia)), consisting of 1000 parallel glass capillaries with a length of 20 cm. The column was maintained at a constant temperature of 30 °C for 15 min. Within the MCC, the sample’s analytes were separated and then transported to the ion mobility spectrometer for detection and quantification. In the spectrometer, the analytes were introduced into the ionization chamber, where they were ionized by a Tritium source with 450 MBq of radioactive activity and an energy of 6.5 keV. Subsequently, the ions were introduced to the drift tube (6 cm of length) where they traveled under a constant electrical field (350 V·cm^−1^) and constant temperature (60 °C) to reach the detector.

Software LAV version 1.5.2 beta from G.A.S was used for data acquisition, collecting spectra in positive ion mode. Parameters were set as follows: each spectrum was formed with an average of 32 scans, the grid pulse width at 100 μs, a trigger repetition period of 21 ms, and the sampling frequency at 150 kHz. More information on the sample protocol can be found in Ref. [55].

Figure 1 presents the raw signal obtained from the MCC-IMS system during the analysis of an oil sample. The output of the MCC-IMS is typically represented with 2D maps: the captured string is reordered in such a way that drift and retention times are represented in the axes of the plot. Each spectrum consisted of 3000 data points, and each collected sample included 1300 spectra. MCC-IMS samples are intricate, and data are highly correlated. Figure 1 also shows the complexity of the MCC-IMS samples and the prominent RIP at approximately 6 ms drift time. The exact position of the RIP changes from sample to sample due to instrument and sampling variability.

### 2.2. Methods

Figure 2 outlines the MCC-IMS data processing workflow presented in this article. The workflow is implemented using the PLS Toolbox from Eigenvector Research (version 9.3), which operates within the MATLAB environment (version 2024b). The workflow begins with the acquisition of MCC-IMS e-nose data along with associated metadata. To enhance signal quality, the raw MCC-IMS data undergo preprocessing, which includes noise reduction, baseline correction, and peak alignment. The effectiveness of the preprocessing workflow in terms of parsimony does not rely crucially on the specific strategy or algorithmic solution of any single step in the pipeline. Rather, it arises from the integration of all preprocessing stages, which collectively contribute to the development of parsimonious models.

Once preprocessed, classification models are developed based on the predefined labels and characteristics of each sample. The models’ accuracy is then evaluated through both internal and external validation.

Noise in the system, minor variations in the experimental protocol, and instrument variability can introduce unwanted fluctuations in the acquired spectra, even for samples of the same type. Therefore, MCC-IMS data must be preprocessed to improve the signal-to-noise ratio and ensure the development of more robust multivariate calibration models.

Noise Reduction

Acquired spectra appear corrupted with random noise that is introduced in the system by uncontrolled sources such as electromagnetic interferences, the thermal noise of the electronic components, acoustic noise due to vibrations of the electrostatic shield at the IMS detector, atmospheric fluctuations, or defects in the components of the system.

Digital filters are usually applied to the original signal to reduce noise while keeping the frequencies that build the signal of interest. Since MCC-IMS data are classified according to the position (time) of the peaks, this position needs to be preserved, avoiding any time shift in the acquired spectra. Therefore, centered non-recursive filters are especially suited for noise reduction in IMS data. Common non-recursive filters used for signal processing include moving average, median, derivative, Whittaker, Savitzky–Golay, and adaptive filters. Savitzky–Golay filters, which use polynomial least squares fitting, and wavelet-based filters, which rely on multiresolution signal decomposition, are both effective at smoothing data while preserving peak shape and signal height. While other filtering strategies may be more effective at reducing noise, they often fail to maintain peak integrity. For this reason, and due to their straightforward implementation, Savitzky–Golay filters are particularly popular in analytical chemistry [56,57,58] and are the method of choice in our workflow for filtering signals with sharp peaks, such as those produced by IMS instruments. While wavelet-based filters are also a viable option, they were ultimately not preferred due to their more complex implementation, although comparable results are expected.

For the optimal performance of the Savitzky–Golay filters, window length and polynomial degree must be fine-tuned [59,60]. Other effective denoising techniques, such as wavelet shrinkage [61,62] and roughness penalty methods [63,64], have shown promise but remain underutilized in ion mobility spectra processing. In this example data, a second-order Savitzky–Golay filter with a window size of 9 points was applied to the acquired spectra. The filter window was selected based on the width of the peaks in the mobility spectra (approximately 100 data points). The polynomial order of the filter was chosen to accurately reconstruct the original signal while avoiding the modeling of noise.

Baseline correction

Baseline correction is used to counteract interferent effects, such as low-frequency background features or residual tails of signal peaks in spectra, which produce undesirable non-zero signals when and where no compounds have been detected.

For 1D signals, a wide variety of numerical techniques can be used for baseline correction such as polynomial fitting, weighted least squares, or methods based on wavelets [65,66,67,68,69,70]. They are based on building a function that effectively behaves as a new baseline. The corrected signal emerges from subtracting the new baseline from the acquired signal, and it should be close to zero at zones without signal peaks. Polynomial fitting is an effective method to estimate baselines but requires user intervention. We considered asymmetric least squares (AsLS) due to its simplicity, fast convergence, and reliable performance in previous implementations for chromatography data [71]. Refinements of the AsLS method can be found in the literature as air PLS [68] and psalsa [72], which can outperform AsLS when high intense peaks compared to baseline are found in the signals. Wavelet methods can also be a useful tool for baseline removal, but, as with the AsLS, the user needs to carefully choose adjustable parameters to achieve effective results.

AsLS minimizes a cost function S that balances two competing terms:(1)$$S=\sum_{i}w_{i}{\left(y_{i}-b_{i}\right)}^{2}+\lambda \sum_{i}{\left(\Delta^{2}b_{i}\right)}^{2}$$
where $y_{i}$ represents the signal and $b_{i}$ is the baseline estimation. The first term in *S* ensures fidelity from the baseline estimation to the signal. The weights $w_{i}$ are defined based on a parameter *p* as follows:(2)$$w_{i}=\left\{\begin{array}{ll} p & if\;y_{i}>b_{i}\\ 1-p & \;if\;y_{i}<b_{i} \end{array}\right.$$

The parameter *p* controls the asymmetry between positive and negative residuals, where *p* = 0.5 results in Ordinary Least Squares [43]. The second term in S penalizes the non-smoothness of $b_{i}$, and it is controlled by the parameter λ. A logarithmic search within the defined ranges for *p* and λ provides robust parameter selection, allowing the same set of parameters to be used across measurements, minimizing manual effort [73]. To remove the baseline from the olive oil dataset, we applied AsLS to all captured spectra using these parameters: λ = 10^5^ and *p* = 5 × 10⁻^3^. These parameter values were selected based on the prior literature, practical considerations for the specific dataset, and a brief trial and error procedure. However, it is important to note that neither the specific baseline correction method (e.g., AsLS) nor the exact parameter values (e.g., λ and *p*) significantly impacted the final results. Sensitivity tests for this dataset revealed that adjusting λ and *p* by up to a factor of 5 produces statistically negligible differences in the final olive oil classification accuracy. Similarly, alternative methods like airPLS or psalsa could also be employed without substantially altering the outcome.

Peak alignment

MCC-IMS-based e-nose instruments generate consistent spectra across different sample types. However, environmental factors like temperature and atmospheric pressure can cause peak shifts in spectra. Consequently, peak alignment is necessary before building a multivariate model to obtain reliable results.

Since the RIP consistently appears at the same drift time, it serves as a reliable reference for aligning spectra. A similar approach applies to instruments in negative mode, using the Reactant Ion Negative (RIN) peak. Alternatively, a peak from an internal standard can also be used for alignment. Misalignments in drift time due to variations in temperature and pressure inside the drift tube can be corrected by expressing the spectra in terms of reduced mobility (Ko). It is expected that working in reduced mobility units makes the analysis unsensible to instrument design and to ambient conditions [27,44]. Following the same principle, if spectra are expressed in terms of drift time, a simple multiplicative adjustment of the RIP position relative to a reference point is enough to align all peaks in the spectra. The correction factor in this method is computed as follows:(3)$$k_{c}=\frac{t_{ref}}{t_{RIP}}$$
where t_ref_ is the drift time of the RIP of a selected reference measurement, and t_RIP_ is the drift time of the spectrum to be aligned. Therefore, a separate kc value is computed for each spectrum to be aligned. The product of kc with the vector representing the drift time of the spectrum yields a corrected drift time axis. Consequently, t_RIP_ is calculated for each spectrum. Further misalignments can be addressed using common warping techniques such as Correlation Optimized Warping (COW), Parametric Time Warping (PTW), and Icoshift. Generally, correlation-based alignment methods like COW tend to outperform distance-based methods like [74,75]. After alignment, linear interpolation is typically required to ensure uniform drift time sampling across all spectra.

Misalignments in retention time can also be important and require correction. Common warping methods, already mentioned above, are often used to address these distortions. Alternative approaches, such as linear regression and spline-based methods, are also available in the literature [73,74]. In our specific olive oil dataset, peak position correction in the spectra was achieved by applying a multiplicative adjustment to the RIP position, using a reference RIP position of t_ref_ = 6.488 ms. Peak alignment along the retention time axis was accomplished by shifting all chromatograms to align with the injection point.

### 2.3. Methods for Data Analysis

After completing the preprocessing steps, the approach used to analyze the cleaned MCC-IMS-based e-nose dataset depends on the specific problem at hand. For targeted analysis, which focuses on identifying and quantifying known compounds, peak picking strategies and regression tools are typically employed. In contrast, untargeted analysis aims to establish unique sample fingerprints, relying on pattern recognition algorithms and classification tools. This work, inspired by the concept of the electronic nose, emphasizes untargeted strategies. These strategies can be divided into two main categories: unsupervised learning algorithms, which extract information from the data structure, and supervised classification algorithms, which use additional information, such as class labels, to classify the data. There is abundant literature and comprehensive studies that thoroughly cover both approaches [76,77,78,79].

#### 2.3.1. Model Selection

Among the many available unsupervised techniques, Principal Component Analysis (PCA) and Parallel Factor Analysis (PARAFAC) are commonly used, while in supervised methods, popular choices include k-nearest Neighbors (KNNs), Linear Discriminant Analysis (LDA), Partial Least Squares Discriminant Analysis (PLS-DA), Artificial Neural Networks (ANNs), and Support Vector Machines (SVMs). In spectral data analysis, it is common to encounter more features than observations, which often leads to overfitting and overly optimistic results in supervised models. However, PLS-DA tends to be less susceptible to this issue compared to other algorithms [80]. For this reason, we chose to develop classification models using the PLS-DA supervised technique in this study.

#### 2.3.2. Evaluation of Model Performance

Validation is a fundamental tool to prevent overfitting in supervised classification techniques. The usual approaches to model validation are the holdout method, resampling, and three-way split methods. The holdout method involves splitting the dataset into two subsets: one for training the model (training set) and another for validation (validation set). While simple, it is the least reliable. Resampling methods, such as bootstrap and cross-validation (including variants like random subsampling, k-fold, and the leave-one-out method), also use a two-way split but repeat the process multiple times, improving reliability. The most reliable approach is the three-way split method, which is used when both model parameter tuning and error estimation must be performed simultaneously. This method divides the dataset into three subsets: training, validation, and test sets. The training set is used to develop prediction models, and the validation set evaluates their performance. This process is repeated until the best model is identified, which is then tested on the test set. A usual strategy to estimate model performance is averaging results from several cross-validation iterations, which assumes that trained models will perform similarly and capture the same information. While this holds true with large datasets, it may lead to incorrect interpretations if these assumptions are not met. This issue is highlighted in the work of Rodríguez-Pérez et al. for PLS-DA data models [81].

We propose using a three-way split procedure based on a double cross-validation method, as shown in Figure 3. In our approach, the complete dataset was randomly divided into a training set (126 samples) and a test set (54 samples). Internal validation was conducted using 5-fold cross-validation, while external validation was assessed using the test samples. This process was repeated 10 times to estimate the error of the accuracy. Double cross-validation provides an added safeguard against overfitting by explicitly separating the hyperparameter selection from the model evaluation process, unlike traditional cross-validation, where hyperparameter selection can be influenced by the same dataset used for evaluation. In the PLS-DA model used in our study, double cross-validation was implemented to prevent overfitting and ensure that accuracy remains consistent and stable across different models generated from various data partitions.

## 3. Results and Discussion

In this section, we will showcase the proposed methodology to preprocess MCC-IMS-based e-nose data illustrated with acquired data from the three types of olive oils described in Section 2.1.

### 3.1. Preprocessing

Data preprocessing starts denoising MCC-IMS spectra. Figure 4 shows the effect of filtering a raw spectrum of an olive oil dataset for different window sizes. While increasing the window size can effectively reduce noise [as seen in Figure 4a], excessively large window sizes can distort the peak shapes [Figure 4b]. In this example, applying a second-order Savitzky–Golay filter with a window size of 9 points successfully reduced noise without altering the peak shapes. After eliminating high-frequency noise, the next step is to remove low-frequency noise associated with the baseline. Figure 4c illustrates the baseline correction of a single spectrum following noise reduction. Initially, the peaks are superimposed on a low-frequency signal. The AsLS algorithm (λ = 10^5^ and *p* = 5 × 10⁻^3^) converges rapidly, allowing the background to be extracted from the signal after just five iterations. Different parameter choices in AsLS can lead to varying background estimations, still requiring the supervised verification of the corrected spectra to prevent undesirable effects, such as peak distortion or truncation. Therefore, human supervision remains essential for validating the estimated baseline function and selecting appropriate parameters. However, due to the similarity between spectra, this inspection is typically only required for a few representative measurements. Implementing AsLS in real time for baseline correction during the data acquisition would be highly desirable. However, due to the iterative nature of the algorithm, the mentioned sensitivity of its performance with the λ and *p* parameters, and the significant computational cost involved, real-time implementation poses a challenge. Consequently, AsLS is typically applied after the measurement has been completed. Finally, the last preprocessing step is peak alignment. Figure 4d displays two spectra after noise reduction and baseline correction. Peaks corresponding to the same chemical compounds initially appear at different drift times, as evidenced by the varying positions of the RIP [Figure 4e]. After alignment [see Figure 4e], the RIPs are aligned at the same drift time (tref = 6.488 ms), and similarly, the peaks corresponding to the same compounds are now properly aligned.

### 3.2. Data Analysis

The preprocessed spectra were used to train and test the PLS-DA models as described in Section 2.3.2. To provide further details of the impact of the preprocessing in this dataset, we explored how each of the preprocessing steps affected the classification accuracy of the models. The whole analysis was repeated, removing one of the preprocessing steps on every repetition. Figure 5 presents classification accuracy on internal validation data as a function of the number of latent variables. Accuracy improves with increasing latent variables until reaching a plateau, regardless of the preprocessing approach (accuracy around 85%). Models built from fully preprocessed data or lacking a noise reduction stage achieve this plateau with fewer latent variables and higher accuracy compared to those where peak alignment is omitted. Models missing baseline removal or without any preprocessing show the lowest accuracy and require more latent variables to reach the plateau. The results suggest that baseline correction has the greatest influence on final model accuracy, while noise reduction using Savitzky–Golay (SG) filtering provides only a minimal enhancement to the overall model performance. The peak alignment stage also contributes to predictive performance, though only to a limited extent. Figure 5 demonstrates that models built without preprocessed data require more latent variables to achieve comparable accuracies, and these models are more dependent on the specific set of measurements used for training. The difference in complexity between models built from fully preprocessed data and those built from raw data are approximately five latent variables. Since target models should be parsimonious, generalizable, and not dependent on the training data, the advantages of the presented approach are evident.

The PLS-DA models built from fully preprocessed data achieved an accuracy of 85% ± 5% in internal validation and 83% ± 5% in external validation. Notably, these results align with previously published studies on olive oil classification (LOO, VOO, EVO). Given the current state of the technology, this result appears to represent the best performance available. It is important to note that simplifying the problem into a two-class classification problem (LOO vs. VOO and EVO) would yield accuracies exceeding 95% [82]. However, the three-classes problem is more challenging due to the similarity between VOO and EVO. Table 1 summarizes the prior contributions, demonstrating consistency despite differences in the number of measured samples and in signal processing and classification methodologies used, such as LDA, tandem PCA-LDA, or OPLS-DA [55,82,83,84,85,86,87].

Generally, PLS-DA models outperform LDA and tandem PCA-LDA approaches. LDA models are prone to overfitting, often yielding overoptimistic results if a rigorous validation procedure is not implemented. Moreover, PCA-based feature extraction, being an unsupervised approach, may incorporate information unrelated to olive oil sample categories, introducing noise into the subsequent LDA supervised stage.

The similarity in final accuracy values reported in Table 1 supports the validity of our current final findings. Furthermore, an additional advantage of using PLS-DA models is their capacity to facilitate the interpretation of results beyond the mere accuracy achieved in external validation, as they include a feature extraction stage prior to classification. Figure 6 reduces the data to two latent variables, with the directions derived from training samples (the first partition of the external validation).

When test samples are overlaid, the model shows good generalization, as both training and test sets cover the same regions in the latent space for different types of olive oils. LOO samples are more distinctly separated from the others, suggesting easier identification compared to EVOO and VOO. This is particularly important since, unlike the other two types, LOO is not certified for human consumption and has lower market value. The results are consistent with those reported by Valli et al. [82], who also used the PLS-DA classifier to differentiate the same olive oil varieties. In their study, the researchers developed PLS-DA models for binary classification tasks, distinguishing one class from the other two. This approach led to four models: EVOO vs. non-EVOO, VOO vs. LOO, LOO vs. non-LOO, and EVOO vs. VOO. The findings indicated that LOO samples showed greater differences compared to EVOO and VOO samples. From a chemical perspective, the easier identification of LOO samples, as supported by the specific literature, is due to their higher levels of oxidation and impurities. In contrast, EVOO and VOO, being of higher quality, exhibit lower degrees of degradation [88].

Lastly, an analysis of the Variable Importance in Projection (VIP) scores from the PLS-DA models reveals regions within the MCC-IMS samples that are relevant for accurate classification. To do this, the VIP scores are calculated across all the models in external validation. The VIP scores were then averaged across models and reshaped to create a matrix that matches the dimensions of the original MCC-IMS e-nose data. The result is visually represented in Figure 7. A feature with a VIP score higher than one is considered important in the model and colored in red hues in the image. These features tend to cluster in 2D peaks, which are associated with the molecular ions responsible for olive oil separation.

## 4. Conclusions

MCC-IMS based e-nose instruments represent an alternative to traditional gas sensor-based e-noses with improved capabilities in selectivity and quantitative measurements. The data analysis of the MCC-IMS measurements is an open field that ranges from preprocessing to data modeling and validation. We have proposed a methodology for the untargeted analysis of MCC-IMS samples using multivariate methods that does not depend on peak integration techniques. The methodology has a strict validation scheme to prevent overfitting the PLS-DA model to the calibration data, to optimize the model meta-parameters and to estimate the performance of the classification.

We have applied this methodology to a specific olive oil analysis dataset, exploring the impact of each of the preprocessing steps on the final classification. Data preprocessing reduced the model’s complexity by lowering the number of latent variables required to achieve stable accuracy during internal validation from ten to five latent variables. In this regard, the baseline correction stage and, to a lesser extent, the alignment stage were more relevant in enhancing model accuracy. The training models built from fully preprocessed data were tested for unseen olive oil samples, obtaining an accuracy of 83% ± 5%.

While achieving comparable results, our work emphasizes key strategies to ensure the robustness of the findings. The preprocessing stage plays a crucial role in ensuring good performance and parsimonious models. To prevent overfitting and overoptimistic results, rigorous internal and external validation are essential. Streamlining the workflow—from raw data to final classification—facilitates these validation procedures and strengthens the reliability of the results.

We do believe that future work can be oriented in improving both the preprocessing, especially baseline estimation, alignment, and model validation, to offer more reliable methods less dependent on manual supervision.

## Figures and Tables

**Figure 1 sensors-25-00737-f001:**
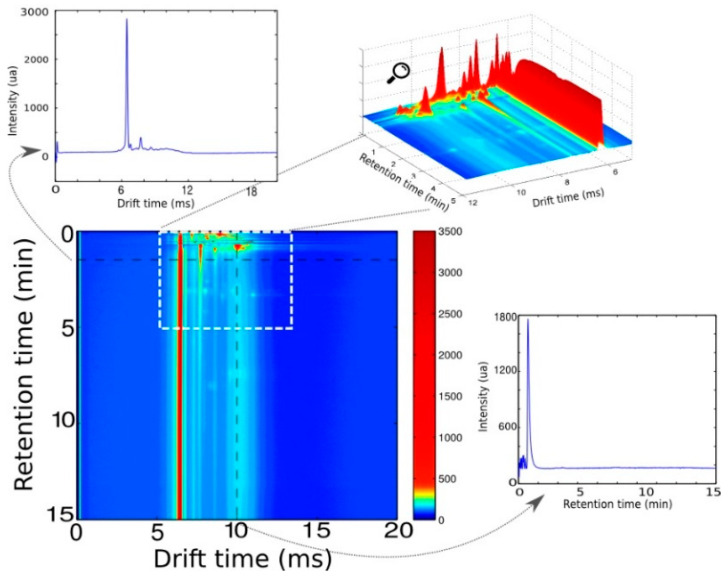
MCC-IMS data acquired from an olive oil sample. An MCC chromatogram and an IMS spectrum are also shown. IMS spectra show prominent peak (RIP) close to 6 ms. The region of the image in which most of the peaks appear is also shown in a three-dimensional plot, showing the complexity of the captured data. Note the non-uniform color scale to highlight the peaks in data.

**Figure 2 sensors-25-00737-f002:**
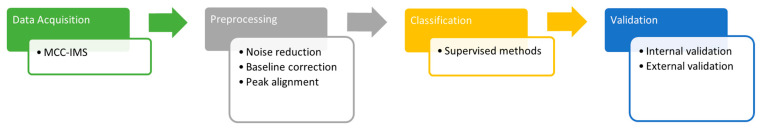
Steps involved in the development of calibration models for MCC-IMS data 2.2.1 preprocessing.

**Figure 3 sensors-25-00737-f003:**
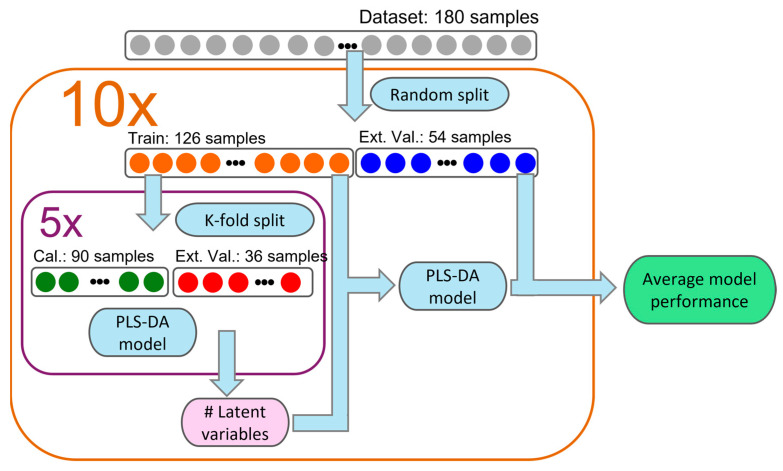
Double cross-validation scheme utilized to evaluate the classification performance of models.

**Figure 4 sensors-25-00737-f004:**
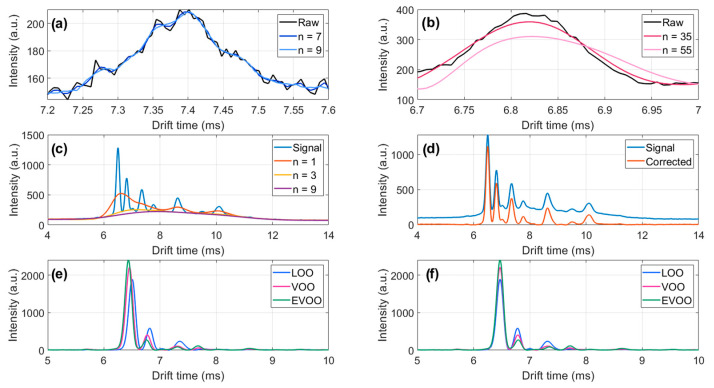
(**a**) Segment of a spectrum before and after applying a second derivative Savitzky–Golay filter with window sizes of n = 7 and n = 9; (**b**) a different segment of the same spectrum filtered with window sizes of n = 35 and n = 39. Note the presence of an optimal window size that minimizes noise while preserving the spectral shape; (**c**) filtered spectrum and baseline estimation using AsLS after various iterations, showing rapid convergence toward accurate baseline estimation; (**d**) filtered spectrum and the resulting spectrum after baseline correction; (**e**) three spectra (acquired at tret = 104 s), each corresponding to one of the olive oil classes (LOO, VOO, and EVOO) after noise removal and baseline correction, demonstrating misaligned peaks; (**f**) the same spectra after peak alignment.

**Figure 5 sensors-25-00737-f005:**
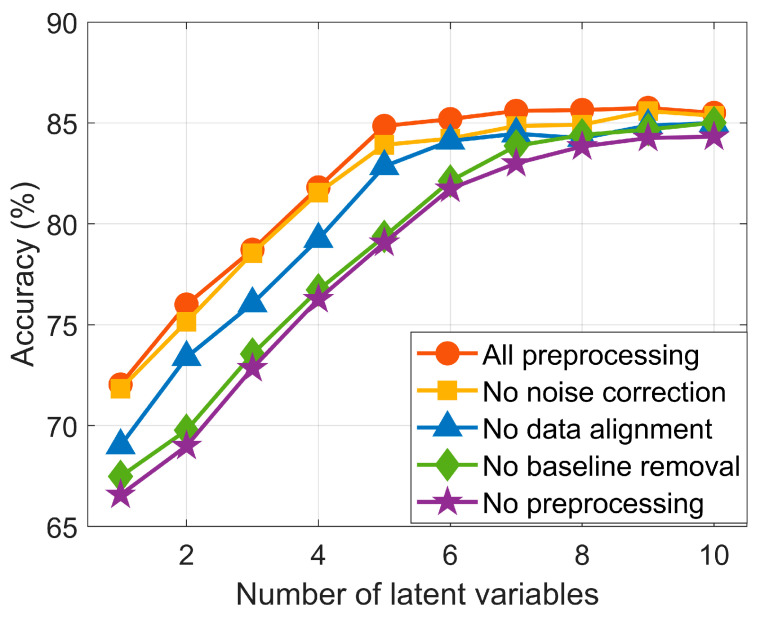
The selection of latent variables was based on optimizing classification accuracy during internal validation. The figure indicates that data preprocessing reduces model complexity while maintaining performance. Baseline removal followed by peak alignment are the preprocessing steps that contribute most to this improvement.

**Figure 6 sensors-25-00737-f006:**
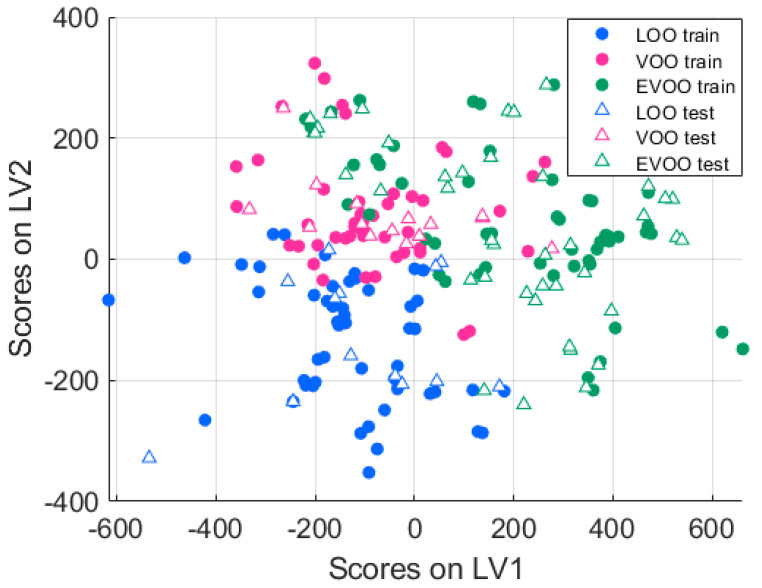
Scores for the first two latent variables of the training set. The same projection is used for the test samples. EVOO samples tend to exhibit higher scores on LV1.

**Figure 7 sensors-25-00737-f007:**
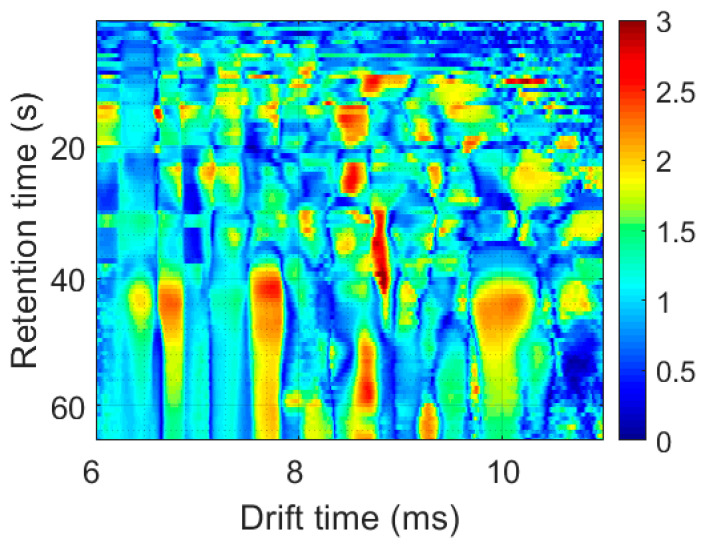
Average VIP scores of the final PLS-DA models. Relevant features for samples’ class separation (VIP score higher than 1) are colored in red hues.

**Table 1 sensors-25-00737-t001:** Summary of information and accuracies achieved in previous studies on olive oil classification (LOO, VOO, EVO).

Reference	Type of Separation	Number of Samples	Accuracy (%)	Chemometric Approach
[55]	MCC	98	87	PCA-LDA
[84]	MCC/GC	55	79/83	PCA-LDA
[83]	GC	701	79	PCA-LDA
[85]	GC	268	94	OPLS
[82]	GC	198	67–95 *	PLS-DA
[86]	GC	120	72–87 *	PCA-LDA
[87]	GC	94	83	PCA-LDA

* A range of accuracies is obtained due to the use of different classification models and methods.

## Data Availability

Dataset available on request from the authors.

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
