# Peer review of "Signal Preprocessing in Instrument-Based Electronic Noses Leads to Parsimonious Predictive Models: Application to Olive Oil Quality Control"

_sensors, 2025, doi:10.3390/s25030737_

Round 1
Reviewer 1 Report
Comments and Suggestions for Authors
In their manuscript, L. Fernandes et al. report on advancing the data preprocessing techniques for the e-nose identification framework and their testing based on the olive oil dataset collected by a multi-capillary column ion mobility spectrometer-based e-nose. The performed study is of high practical interest as e-nose systems are actively developing nowadays and being integrated into many fields, from production quality control and monitoring air quality to screening human diseases. Managing large quantities of the data being recorded in real-time is one of the key issues to be solved for the beneficial application of e-nose units in these fields. The work is done on a decent level with a clear, thorough, and scientifically sound flow of the text. The manuscript is recommended to be published in the Sensor journal after Minor revision according to the queries below:
1. Noise Reduction: it is necessary to provide the total number of data points for each measurement for which the window size of 9 points for the Savitzky-Golay filter was employed for clearness. Since for the narrow peaks such a window width can be too high, inappropriately broadening it and cutting its intensity too much
2. Baseline correction. For the described AsLS algorithm it would be convenient to add the corresponding relation since its parameters (lambda and p) are described.
Also, please clarify, can the AsLS algorithm be integrated to the e-nose software to e applied automatically, during the measurements, not manually by processing the data after the measurements?
3. Figure 3. What are the factors 10x and 5x in the boxes? it is not evident which parameter they are related to
4. Identification accuracy of ca. 83% is not high enough for practical applications. Although it is not in the scope of the work, authors are encouraged to provide their suggestions on which factors could improve the accuracy to >95% and what strategies can be proposed to achieve these results
Author Response
|
Comments 1: Noise Reduction: it is necessary to provide the total number of data points for each measurement for which the window size of 9 points for the Savitzky-Golay filter was employed for clearness. Since for the narrow peaks such a window width can be too high, inappropriately broadening it and cutting its intensity too much
|
|
Response 1: Thank you for your valuable feedback, which has allowed us to clarify and improve this aspect of our manuscript. The Savitzky-Golay filter is well-suited for smoothing noisy signals while preserving peak shape. However, we agree that the relationship between the window size and the number of data points is critical for optimal performance.
In our case, each spectrum consisted of 3,000 data points, and each collected sample included 1,300 spectra. The selected window size was 9 points, while the peaks in our data typically span more than 100 data points. This ensures that the filter effectively smooths the data without distorting the peak shape. To illustrate this, an example of the filter's performance is provided in Figures 4a and 4b, where the integrity of the peaks is visibly preserved when using a small window size.
In response to your suggestion, the manuscript has been updated to provide this information, and changes can be found between lines around 307 and 312. Additional information is provided at line 255. These updates have been highlighted in yellow in the revised manuscript |
|
Comments 2 Baseline correction. For the described AsLS algorithm it would be convenient to add the corresponding relation since its parameters (lambda and p) are described.
|
|
Response 2: Thank you for your valuable suggestion. We agree that including the equations for the AsLS algorithm would be beneficial to illustrate the relationship between its parameters. In response to your comment, we have added the corresponding equations to the revised manuscript, which can be found between lines around 329 and 337. The changes are highlighted in yellow.
Comments 3: Also, please clarify, can the AsLS algorithm be integrated to the e-nose software to e applied automatically, during the measurements, not manually by processing the data after the measurements?
Response 3: Thank you for your insightful comment. Indeed, real-time baseline correction would be highly desirable. However, the iterative nature of the algorithm and its computational cost present significant challenges, which we did not address in this work. In our study, we have applied the AsLS algorithm in a semi-automated fashion, as is commonly done, at the end of the measurements, requiring some parameter exploration and tuning.
In response to your comment, we have added a brief paragraph between around lines 456 and 461. All changes are highlighted in yellow in the updated manuscript.
Comments 4: Figure 3. What are the factors 10x and 5x in the boxes? it is not evident which parameter they are related to. Response 4: Thank you for your question. The explanation was originally provided in the figure captions, but we agree that the manuscript is improved by having this explanation in the main text. Therefore, we have moved and rearranged the content to better clarify this point.
The changes have been incorporated into the manuscript between around lines 423 and 432, and are highlighted in yellow for your convenience.
Comments 5: Identification accuracy of ca. 83% is not high enough for practical applications. Although it is not in the scope of the work, authors are encouraged to provide their suggestions on which factors could improve the accuracy to >95% and what strategies can be proposed to achieve these results Response 5: Thank you for your valuable feedback. We acknowledge that an accuracy of 83% may be considered at the threshold for practical applications, but it is consistent with other studies in the literature. It is important to note that the classification problem in our work involves three classes, two of which (EVOO and VOO) are quite similar. Some studies reduce the problem to a binary classification (LOO vs EVOO and VOO), achieving better performance, but for three classes, our result aligns with the current state of the art.
However, our contribution demonstrates that, with preprocessing and proper validation, the accuracy remains stable across different datasets, and the models maintain parsimony. We believe that further improvements could be achieved by including a feature selection process, exploring more advanced classification algorithms, or enhancing data preprocessing steps.
Changes addressing your comment can be found between around lines 534 and 543. All changes are highlighted in yellow for your convenience.
|

Reviewer 2 Report
Comments and Suggestions for Authors
The paper presents an interesting research work. However, it can still be improved before final publication.
First, the title seems strange, with even a dot at the end. is this a mistake?
Second, the references are too many and too old. I recommend replacing some papers on electronic noses with papers published after 2023.
Forth, Paper 80 also mentions PLS-DA, what is the clear difference and improvment of this work compared to paper 80? Please add more comparison and analysis.
All in all, the paper can be greated improvement by improving the references and experimental design.
Third, the papers directly use S-G filter instead of the other one, which makes the superiority of S-G blur. Please compare at least another three methods, such as adaptive filter, Whittaker,and derivatives.
Author Response
|
Comments 1: First, the title seems strange, with even a dot at the end. is this a mistake?
|
|
Response 1: Thank you for your feedback. In response to your suggestion, we have revised the title of the manuscript to better reflect the scope and essence of the work. The new title is: Signal preprocessing in instrument-based electronic noses leads to parsimonious predictive models: application to olive oil quality control. We hope the new title aligns more closely with the intent and focus of the article. Thank you again for your constructive comment. Change is highlighted in yellow at the title section. |
|
Comments 2: the references are too many and too old. I recommend replacing some papers on electronic noses with papers published after 2023
|
|
Response 2: Thank you for your valuable feedback. We acknowledge that the reference list is extensive, as we aimed to provide a comprehensive background on the topic. However, we recognize the importance of including recent studies to ensure the relevance of the cited works. In response, we have updated the bibliography to incorporate recent papers, most of the published after 2023, particularly in the field of electronic noses, replacing some older references where appropriate. Nevertheless, certain foundational works have been retained as they remain critical to understanding the development and context of this research area. Changes can be found at the reference section highlighted in yellow in the upgraded manuscript.
Comments 3: the papers directly use S-G filter instead of the other one, which makes the superiority of S-G blur. Please compare at least another three methods, such as adaptive filter, Whittaker,and derivatives.
Response 3: Thank you again for your suggestion. The specific filtering method, as long as it is suitable, does not significantly affect the final result. Savitzky-Golay and wavelet-based filters are both appropriate for smoothing noisy signals while preserving peak shape, which is particularly desirable for our type of data. We selected Savitzky-Golay due to its simpler implementation. Wavelet-based filters are also a viable option but, the specific method employed in each preprocessing step is not critical. What truly matters is the integration of the entire workflow, ensuring that suitable methods are applied at each stage.
Other smoothing methods, such as moving average or median filters, although more efficient for noise reduction, fail to preserve peak shape and are therefore unsuitable for our purposes. Furthermore, the use of wavelet-based filters for noise reduction, or alternative strategies for baseline correction, or for peak alignment, yields statistically comparable results in terms of accuracy, accuracy stability, and parsimony to those reported.
In response to your request and with the aim of clarifying this issue, we have included a paragraph between lines around 295–304. All changes are highlighted in yellow in the updated manuscript. Additionally, the general concept regarding the relative importance of specific methods versus the holistic application of the workflow has been introduced between lines 270–274.
Comments 4: Paper 80 also mentions PLS-DA, what is the clear difference and improvement of this work compared to paper 80? Please add more comparison and analysis. Response 4: Thank you for your thoughtful feedback. First, we would like to clarify that the reference previously mentioned as [80] in the original manuscript is now updated to [85] due to changes and additions in the bibliography.
Regarding the comparison with the work by Valli et al. (reference [85] in the updated manuscript), while their study indeed employs PLS-DA, it does not implement double cross-validation. This lack of rigorous validation leads to models with broader variability in stability, as evidenced by the wide range of accuracies reported in their results table. Furthermore, in their work, the best performance is achieved by focusing on a simplified binary classification task: separating LOO from the combined EVOO and VOO classes. This binary problem is inherently less complex and therefore yields higher accuracies.
In contrast, our study addresses a more challenging three-class classification problem, achieving results comparable to other literature but with significantly more stable accuracies across validation iterations. Additionally, our approach emphasizes parsimony, delivering robust models with fewer latent variables while maintaining high classification performance.
We hope this clarifies the differences and improvements in our work. These details have been incorporated into the manuscript, with updates located between lines 505–510 and 534–543, highlighted in yellow for ease of reference.
|

Reviewer 3 Report
Comments and Suggestions for Authors
This study presents a robust methodology for preprocessing MCC-IMS-based e-nose data to classify olive oils (LOO, VOO, EVOO) effectively, achieving 85% internal validation accuracy and 83% external validation accuracy using PLS-DA models. The preprocessing steps, including noise reduction, baseline correction, and peak alignment, significantly enhance model performance and generalization, highlighting the importance of systematic data preparation in e-nose applications.
Abstract:
I. The abstract should explicitly state the precise value of research improvement.
Introduction:
II. How do the challenges of data preprocessing (e.g., RIP tail correction, noise reduction, baseline correction) compare to those faced by traditional e-nose systems? Are these challenges unique to MCC-IMS or GC-IMS?
III. Transition more smoothly between the historical development of e-noses and the discussion on MCC-IMS and GC-IMS technologies. This could be done by summarizing the key limitations of traditional systems before introducing next-generation solutions.
IV. While several application areas are mentioned, consider providing a brief example or statistic to substantiate the claims, particularly for emerging fields like disease diagnosis or environmental monitoring.
V. Refer to new references on e-nose such as DOI: 10.1016/j.snb.2022.131418.
Materials and Methods:
I. Does this temperature minimize oxidation or other degradation processes in the oil?
II. The parameters for AsLS (e.g., λ = 10⁵, p = 5 × 10⁻³) are specified. Could you describe how these values were chosen and whether the performance was compared with alternative methods like air PLS or psalsa?
III. Include a comparison of preprocessing techniques (e.g., Savitzky-Golay vs. wavelet shrinkage) to support the choice of methods used in this study.
IV. Humidity is an important factor in the electrical conductivity of gas sensors. (Look at this article: DOI: 10.1109/JSEN.2020.3038647). It is necessary to discuss the effect of humidity on electrical resistance in the introduction or refer to the relevant reference in the manuscript.
Results and Discussion:
I. Specify the rationale for choosing a second-order Savitzky-Golay filter over alternative methods. Highlight why a window size of 9 points was optimal for this dataset.
II. Discuss whether alternative algorithms for baseline correction were considered and justify the choice of AsLS.
III. Discuss any limitations of the PLS-DA model, such as overfitting, and suggest steps taken to mitigate these issues.
IV. Expand on why LOO samples are more easily identified than EVOO and VOO. Is this due to specific chemical signatures or differences in spectral patterns?
Author Response
|
Comments 1: Abstract. The abstract should explicitly state the precise value of research improvement.
|
|
Response 1: We agree with this comment; the abstract in the original manuscript was indeed too vague. To address this, we have updated the abstract to explicitly highlight the key result: the demonstration that preprocessing workflows mitigate overfitting and produce parsimonious models while maintaining accuracy. Additionally, we have revised the title of the manuscript to better emphasize this central idea. The changes are highlighted in yellow in the revised manuscript and can be found between lines around 29 and 35 on page 1.
|
|
Comments 2: Introduction. How do the challenges of data preprocessing (e.g., RIP tail correction, noise reduction, baseline correction) compare to those faced by traditional e-nose systems? Are these challenges unique to MCC-IMS or GC-IMS?
|
|
Response 2: This is an important issue that was not clearly emphasized in the original manuscript. We have, accordingly, modified the introduction to emphasize this point. We have added a specific paragraph explaining some commonalities and some differences between preprocessing of gas-sensor based enoses and instrument based enoses. Changes can be found on page 2, around lines around 71-87. The revised text is highlighted in yellow in the updated manuscript.
Comments 3: Introduction. Transition more smoothly between the historical development of e-noses and the discussion on MCC-IMS and GC-IMS technologies. This could be done by summarizing the key limitations of traditional systems before introducing next-generation solutions.
Response 3: We agree with your comment. To address it, we have expanded the original paragraph in the manuscript to introduce key limitations of traditional systems, such as limited selectivity and detection limits. These limitations justify the need for new approaches, even though these approaches come with their own challenges. Additionally, we have also added and updated the references. The revised text can be found between lines 55 and 66 in the updated manuscript and it is highlighted in yellow.
Comments 4: Introduction. While several application areas are mentioned, consider providing a brief example or statistic to substantiate the claims, particularly for emerging fields like disease diagnosis or environmental monitoring.
Response 4: Thank you for your comment. Certainly, adding examples to the applications areas improves the manuscript. We have expanded the original paragraph in the manuscript to introduce examples for diverse e-nose applications. The expanded text can be found on page 2, between lines around 45 and 55 in the updated manuscript and it is highlighted in yellow.
Comments 5: Introduction. Refer to new references on e-nose such as DOI: 10.1016/j.snb.2022.131418
Response 5: The new reference has been introduced. It is reference number 11 and it is relevant in the explanation about the transition of traditional sensor-based electronic noses and next generation instrument-based electronic noses. The reference is located at line 61 on page 2 and in the references section.
Comments 6: Materials and methods. Does this temperature minimize oxidation or other degradation processes in the oil?
Response 6: Thank you for your excellent question. The temperature and incubation time were carefully selected to optimize aroma extraction from the oils while avoiding sample degradation. Published studies indicate that degradation at 60°C becomes significant only over extended periods (in the order of one day). In contrast, the 10-minute incubation period used in our study is negligible in terms of inducing significant degradation but facilitates the transfer of volatiles to the headspace. To clarify this point, we have added a specific paragraph and relevant references to the manuscript. The new paragraph and references can be located between lines 233 and 236.
Comments 7: Materials and methods. The parameters for AsLS (e.g., λ = 10⁵, p = 5 × 10⁻³) are specified. Could you describe how these values were chosen and whether the performance was compared with alternative methods like air PLS or psalsa?
Response 7: Thank you for the opportunity to clarify this issue. The parameters were selected based on prior literature, practical considerations, and a trial-and-error process. Regarding the specific values of the parameters, it is important to note that there is a wide range of values for which the results remain nearly identical. For example, changing lambda by a factor of 5 from the selected value does not significantly affect the baseline estimation. On the other hand, airPLS and psalsa are refinements of AsLS. While they can also be used, they do not substantially improve the accuracy of the results or the parsimony of the model. This principle generally applies to every step of the workflow: the specific method chosen for each step is not critical, as long as it is suitable. Different appropriate strategies will yield comparable results. We have added a paragraph at the beginning of the Methods section (lines 271–275) emphasizing this general concept. In response to your valuable question regarding AsLS, we have included a paragraph between lines around 342–349. All changes are highlighted in yellow in the updated manuscript.
Comments 8: Materials and methods. Include a comparison of preprocessing techniques (e.g., Savitzky-Golay vs. wavelet shrinkage) to support the choice of methods used in this study.
Response 8: Thank you again for your suggestion. In line with our response to comment 7, the specific filtering method, as long as it is suitable, does not significantly affect the final result. Both Savitzky-Golay and wavelet-based filters are appropriate for smoothing noisy signals while preserving peak shape, which is particularly desirable for our type of data. We have chosen Savitzky-Golay due to its simpler implementation. Wavelet-based filters are also a very good option, but again, the specific method used in each preprocessing step is not critical. What truly matters is the combination of the entire workflow (always using suitable methods for each step, of course). Other smoothing methods, such as moving average or median filters, although more efficient for smoothing tasks, do not preserve the shape of the peaks and, as a consequence, are not suitable. In response to your request and in the hope of clarifying this issue, we have included a paragraph between lines 296-305. All changes are highlighted in yellow in the updated manuscript.
Comments 9: Materials and methods. Humidity is an important factor in the electrical conductivity of gas sensors. (Look at this article: DOI: 10.1109/JSEN.2020.3038647). It is necessary to discuss the effect of humidity on electrical resistance in the introduction or refer to the relevant reference in the manuscript. Response 9: Yes, you are right. Humidity is indeed an important factor for gas sensor-based electronic noses, as well as for those based on GC-IMS. We have added a brief comment to highlight this and have included the reference you suggested. The changes can be found between lines around 71 and 87, and the new reference is now reference 21.
Comments 10: Specify the rationale for choosing a second-order Savitzky-Golay filter over alternative methods. Highlight why a window size of 9 points was optimal for this dataset. Response 10: Thank you again for your suggestion. The polynomial order of the filter was chosen to accurately reconstruct the original signal while avoiding the modeling of noise. Each spectrum consisted of 3,000 data points. The filter window was selected based on the width of the peaks in the mobility spectra (approximately 100 data points). Again, a certain range of values could also be used without significantly affecting the final result. For example, an order of 3 or 4 could also work well, but in line with the principle of parsimony, we prefer simpler implementations, as long as they capture the complexity of the data avoiding overfitting. The same can be said about window length. Lengths around 10% of the total number of points would also work well and produce comparable results. Figure 4a and b also point to this fact. A specific paragraph clarifying this issue has been added between lines 309 and 313. The text is highlighted in yellow in the updated manuscript. Comments 11: Discuss whether alternative algorithms for baseline correction were considered and justify the choice of AsLS. Response 11: We appreciate your suggestion. The choice of AsLS was made primarily due to its ease of implementation and rapid convergence. Apologies for being repetitive, but once again, methods such as psalsa are also suitable; however, we did not implement them because they require more complex parameter tuning (psalsa has 3 parameters compared to the 2 in AsLS). Additionally, the final performance, in terms of model stability and parsimony, is not significantly affected by the specific method chosen for each step of the workflow (as long as the selected method is suitable). The changes have been incorporated as mentioned in the response to comment 7 (lines around 342–349). Comments 12: Discuss any limitations of the PLS-DA model, such as overfitting, and suggest steps taken to mitigate these issues. Response 12: Thank you once more for your valuable suggestion. Certainly PLS-DA as a supervised method, tends to overfit. The double-cross validation method implemented is a safeguard to prevent overfitting. A specific paragraph has been added to emphasize the response to your suggestion. It can be found between lines 424 and 433. Comments 13: Expand on why LOO samples are more easily identified than EVOO and VOO. Is this due to specific chemical signatures or differences in spectral patterns? Response 13: Thank you very much for your question. Both aspects are indeed true. There are distinct chemical signatures that lead to differences in spectral patterns. We appreciate the opportunity to elaborate on this in more detail. A specific paragraph addressing your question has been added to the updated manuscript, and the relevant changes can be found between lines around 535 and 544. Additionally, a reference supporting the existence of these different chemical signatures has been included (Reference 88). |
